# Salinity Shapes the Microbial Communities in Surface Sediments of Salt Lakes on the Tibetan Plateau, China

**Yuanyuan He [1,2], Lele He [1,2], Zhe Wang [1,2], Ting Liang [1,2], Shichun Sun [2,3] and Xiaoshou Liu [1,2,*]**

1 Frontiers Science Center for Deep Ocean Multispheres and Earth System, College of Marine Life Sciences, Ocean University of China, Qingdao 266003, China
2 Institute of Evolution and Marine Biodiversity, Ocean University of China, Qingdao 266003, China
3 Fisheries College, Ocean University of China, Qingdao 266003, China
* Correspondence: liuxs@ouc.edu.cn; Tel.: +86-532-82031735

**Abstract:** The extreme geographical and climatic conditions of the Tibetan Plateau result in lakes spanning a diverse range of environmental conditions. Studying microbial response to extreme environmental conditions is important for understanding their adaptation and evolution in the natural environment. In this study, the microbial community composition in the surface sediments from 12 lakes with different salinities on the Tibetan Plateau were analyzed using the Illumina high-throughput sequencing platform. The results showed that the phyla *Proteobacteria* and *Bacteroidota* were the major bacteria, and *Crenarchaeota* was the major group of archaea in low and moderately saline lakes (salinity 0.64–33.88PSU), whereas *Firmicutes* and *Halobacterota* increased significantly in high saline lakes (salinity 44.25–106.9PSU). Salinity was the most important factor impacting bacterial α-diversity, characterized by a significant decrease in microbial diversity indices with increasing salinity. Salinity was also the main driving factor determining the microbial community composition of these lakes. Other factors, including Chl-*a*, OM and glacial meltwater, also had important effects to some extent. In contrast, geographic factors had no remarkable effect on microbial community composition.

**Keywords:** high-throughput sequencing; microbial diversity; salinity; sediment; salt lakes; Tibetan Plateau

## 1. Introduction

Salt lakes are an important part of the global terrestrial aquatic ecosystem, accounting for about half of the world's total inland surface water [1]. They are mainly distributed in arid and semi-arid regions on all continents of the world, such as the rain-shadowed regions of California and Nevada, the Caribbean Plateau of British Columbia (Canada), and the middle belt of the Asia–Africa–European salt lake region (including the Tibetan Plateau in China, the Dead Sea in Palestine and the salt lakes in southwestern Russia). The distribution of salt lakes is also influenced by topography and terrain, ranging from the Dead Sea—the lowest elevation—to the salt lakes of the Tibetan Plateau and South American salt lakes—the highest elevations in the world [2].

Located in southwest China, the Tibetan Plateau is the birthplace of many Asian rivers, therefore, it is called the "Asian Water Tower." It is also the region with the most numerous high-altitude lakes in the world (average elevation exceeds 4000 m) [3]. These lakes span different environmental conditions, from diluted glacier meltwater, to salt lakes and seasonal or perennial snow-covered stratified lakes [4]. Owing to the relatively primitive environmental conditions, they span large geographical distances and are subject to little human interference, making them relatively sensitive to climate and environmental changes. Therefore, the lakes on the plateau represent the most primitive natural environment. Salt lakes are the main type of lakes in the region [5]. Compared with lakes in other regions, lakes on the Tibetan Plateau are characterized by high altitude, low temperature, limited nutrients,

and some lakes are also fed by glaciers and snow melt, which determines the specificity of the environmental proxies [6]. The low air pressure, high altitude, harsh climate and difficult transportation make their study fraught with difficulties. The Qiangtang Plateau is the most typical area on the Tibetan Plateau with the highest altitude. Most of the lakes in the southern Qiangtang Plateau are brackish water with a salinity of 0.26–16.18PSU, and a few are salt lakes. More than 70% of the salt lakes are in its arid north, with salinities ranging from 20 to 100PSU and some even reaching 300PSU [7]. To better understand the effects of salinity and other factors on microbial communities, we selected 12 lakes with a certain salinity range and geographical distance in Naqu, located in the northern part of Qiangtang.

Environmental microbial diversity can reflect the main characteristics of the ecological composition of a community, which is also an essential indicator for assessing environmental conditions [8]. In recent years, interesting studies have been published on salt lakes in Australia, Romania, and other locations in hot and temperate climates. Boggs et al. studied the salt lakes of the northern agricultural region in Western Australia; they found the benthic microbial communities comprised either cohesive to loosely mucilaginous mats, or thin films of diatoms [9]. Eder et al. also found *Halanaerobium* to be a typical species of the Dead Sea, characterized by high salinity and high temperature [10]. Focusing on the special habitat conditions of salt lakes on the Tibetan Plateau, many scholars have conducted studies on the composition and structure of microbial communities of salt lakes, their response to environmental factors, and the metabolic substances and pathways of microorganisms in extreme environments [11–14]. However, most of them focus on individual lakes and the microbial community composition of water, such as Qinghai Lake, Xiaochaidan Salt-Lake [15,16], and there is a lack of holistic study of lakes on the Tibetan Plateau. Therefore, the microbial diversity of high-altitude salt lakes is still poorly understood, and different salt lakes have different physical and chemical properties, which restrict microbial species' diversity and species' resource distribution [17]. Currently, there are various approaches to studying environmental microbial diversity [18]. High-throughput sequencing technology has been frequently used to study microbial diversity in various environments in recent years because of its low cost, high sequencing depth, and easy identification of low-abundance community species [19–23].

The biomass and abundance of microbial groups in the lake sediments are high and have been identified as major reservoirs of carbon and other elements [20]. Microorganisms are the basic driving factors of biogeochemical cycles and have a profound impact on the processes of element decomposition, humus synthesis, material transformation, and energy transfer in sediments [24]. In this study, the microbial community compositions in the surface sediments of 12 Tibetan lakes with different salinity gradients were studied using high-throughput sequencing to deepen the understanding of microbial diversity in the region and provide theoretical support for the comparison of microbial community structure, species diversity, and distribution characteristics in salt lakes in the future. The present study tries to answer the following questions: (1) Is salinity a significant influencing factor in the composition of microbial communities in salt lakes on the Tibetan Plateau? It is assumed that the distribution is based on the salinity gradient. (2) Do other physicochemical factors or spatial factors contribute more to microbial community variation than salinity?

## 2. Materials and Methods

### 2.1. Study Site and Sampling

Most of the lakes studied were located in Naqu (83.91–95.08° E, 29.91–36.5° N) which is located in the northern part of the Tibet Autonomous Region, China, in the hinterland of the Tibetan Plateau. The average altitude is above 4500 m. The region is called the roof ridge of the world and is also the source of the Nujiang, Lhasa, Yigongzangbu, and other rivers [25]. Naqu has a subtropical monsoon climate with low temperature, a lack of oxygen, great difference in temperature between day and night, and windy weather. The average annual temperature is −2.8–1.6 °C, the annual precipitation is 247.3–513.6 mm [26],

and the annual evaporation is 1500–2300 mm, which increases gradually from southeast to northwest [27].

Multiple closed lake systems with different salinities exist in the region. To better understand the environmental factors, including the gradient effect of salinity and the influence of geographical distance on the microbial community composition of lake sediments, a total of 12 lakes were sampled from July to August 2020: Pongcê Co (BZC), Chaxiabu Co (CXBC), Gogen Co (GGC), Bobsêr Co (PSEC), Guojialun Co (GJLC), Dagze Co (DZC), Yangnapeng Co (YNPC), Angdaer Co (ADEC), Bangkog Co (BGC), Gangtang Co (GTC), Yibug Caka (YBCK), and Tangqung Co (DXC) (Figure 1 and Table 1). From each site, the surface sediments (top 5 cm) were collected using a stainless-steel spoon in the coastal zone with an average water depth of less than 0.5 m, and a total of 12 sediment samples were collected. These sediment samples were kept on ice to maintain a low temperature, and quickly transported to the laboratory. One part of each sediment sample taken from each site was frozen in a sterilized 10 mL cryopreservation tube at –80 °C, and an appropriate amount of sediment sample was frozen at −20 °C for the determination of environmental factors and other related parameters.

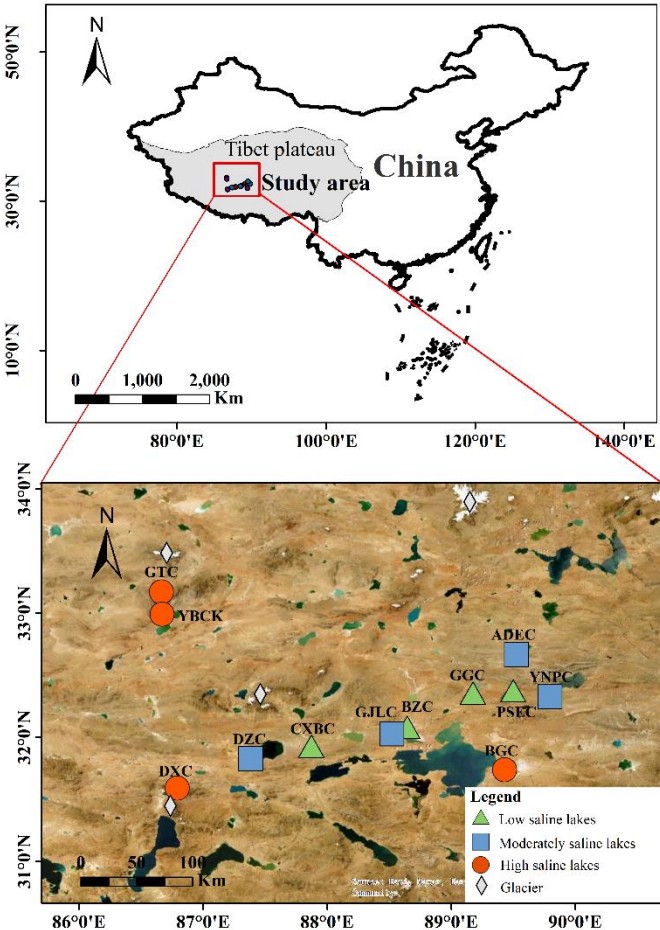

**Figure 1.** Locations of the 12 sampled sites on the Tibetan Plateau, China. The different shapes represent different lake salinity levels, respectively, low (<5PSU), moderately (5–35PSU), and high (>35PSU) saline lakes.

**Table 1.** Geographic parameters and physicochemical characteristics of the 12 sampled lakes on the Tibetan Plateau, China.

| Lake | Longitude (E) | Latitude (N) | Altitude (m) | Distance to Glaciers (km) | S(PSU) | pH | DO (mg/L) | T (°C) | W (%) | OM (%) | Chl-*a* (μg/g) |
|------|---------------|--------------|--------------|---------------------------|--------|-----|-----------|--------|-------|--------|----------------|
| BZC | 88°38′42.50″ | 32°04′57.88″ | 4536 | 81.00 | 0.64 | 8.9 | 6.81 | 20.6 | 15.83 | 3.41 | 0.06 |
| CXBC | 87°52′21.06″ | 31°55′45.66″ | 4499 | 44.00 | 2.89 | 8.96 | 6.42 | 19.18 | 23.57 | 0.33 | 0.16 |
| GGC | 89°10′30.44″ | 32°21′23.87″ | 4675 | 120.00 | 4.67 | 8.98 | 7.26 | 12.74 | 13.59 | 1.60 | 0.67 |
| PSEC | 89°29′58.92″ | 32°22′29.93″ | 4609 | 91.00 | 4.7 | 9.00 | 6.62 | 13.53 | 15.95 | 0.49 | 1.66 |
| GJLC | 88°31′18.15″ | 32°01′40.35″ | 4532 | 83.00 | 10.2 | 8.94 | 5.38 | 16.14 | 15.18 | 0.39 | 0.05 |
| DZC | 87°23′02.71″ | 31°49′40.52″ | 4472 | 28.00 | 11.82 | 8.19 | 6.62 | 18.86 | 21.29 | 1.51 | 0.39 |
| YNPC | 89°47′49″ | 32°19′46.64″ | 4633 | 124.00 | 32.31 | 9.03 | 3.58 | 18.55 | 14.58 | 0.87 | 1.60 |
| ADEC | 89°31′36″ | 32°40′00″ | 4854 | 143.00 | 33.88 | 8.87 | 5.33 | 13.28 | 12.93 | 0.37 | 0.04 |
| BGC | 89°25′49.61″ | 31°44′21.86″ | 4531 | 100.00 | 44.25 | 9.24 | 7.2 | 14.4 | 36.40 | 1.88 | 1.53 |
| GTC | 86°39′51.08″ | 33°10′17.90″ | 4872 | 7.00 | 45.52 | 8.85 | 3.73 | 13.78 | 16.99 | 1.03 | 0.11 |
| YBCK | 86°40′06.09″ | 32°59′31.10″ | 4559 | 14.00 | 49.85 | 8.76 | 5.36 | 16.32 | 58.33 | 12.54 | 1.62 |
| DXC | 86°47′30.24″ | 31°35′26.16″ | 4469 | 6.00 | 106.9 | 8.80 | 12.82 | 15.78 | 10.63 | 0.61 | 0.02 |

Note: Pongcê Co (BZC), Chaxiabu Co (CXBC), Gogen Co (GGC), Bobsêr Co (BSEC), Guojialun Co (GJLC), Dagze Co (DZC), Yangnapeng Co (YNPC), Angdaer Co (ADEC), Bangkog Co (BGC), Gangtang Co (GTC), Yibug Caka (YBCK), Tangqung Co (DXC). S, salinity; T, water temperature; DO, dissolved oxygen; W, water content; OM, organic matter; Chl-*a*, chlorophyll a content.

## 2.2. Determination of Environmental Parameters

The monitoring data for the sediment ecological environment in the study area included salinity (S), oxygen content (DO), temperature (T), pH, organic matter content (OM), chlorophyll a content (Chl-*a*), and water content (W) (the ratio of water weight to total weight of sediment). Based on the described research methods previously [28], the temperature, salinity, oxygen content and pH for the bottom water above surface sediments were measured in situ using a multiparameter water quality analyzer (ProDSS, YSI, USA). The samples were extracted with 90% acetone for 24h and then Chl-*a* was determined by fluorescence spectrophotometry (Trilogy, Turner, USA) [29]. OM measurement was performed using the $K_2Cr_2O_7$-$H_2SO_4$ oxidization method. In the concentrated sulfuric acid, a certain amount of $K_2Cr_2O_7$ was added to convert the organic carbon in the sample to carbon dioxide under heating conditions, and the remaining $K_2Cr_2O_7$ was recovered with ferrous sulfate, and the amount of OM in the sample was calculated according to the amount of $K_2Cr_2O_7$ consumed. Water content was determined by gravimetric method, the wet sediment sample with known weight was dried to constant weight at 105±1 °C, and the water content of the sample was calculated by the difference between the two weights (GB 17378.5-2007).

## 2.3. DNA Extraction, High-throughput Sequencing and Sequence Analysis

Total genomic DNA from the 12 samples was extracted using a DNeasy PowerSoil Kit (Qiagen, USA) according to the manufacturer's instructions. The high-throughput sequencing primer pair 515FmodF (5′ -GTGYCAGCMGCCGCGGTAA-3′) and 806RmodR (5′ -GGACTA CNVGGGTWTCTAAT-3′) was used to amplify the V4 region of the 16S rRNA gene. Equimolar amounts of PCR products were collected and sequenced on a MiSeq PE300 platform (Illumina, USA) at Shanghai Meiji Co., Ltd. (Shanghai, China). Using the FLASH1.2.11 software platform and Fastp0.19.6, the original sequencing data were spliced into two-terminal sequences, and quality control was performed. Sequences with a mean quality score <20 were filtered out. QIIME 1.9 and USEARCH 7.0.1090 were used to analyze the sequencing data. OTU clustering (97% accuracy) was performed on non-repeated sequences (excluding single sequence). Chimeras were removed during the clustering process to obtain representative OTU sequences. Using the RDP Classifier 2.11, the obtained OTU sequences were classified and annotated according to the SILVA 138 database.

*2.4. Data Analysis*

After obtaining the original OTU list, it was flattened according to the minimum sequence number of samples, to avoid errors caused by different sequencing depths in subsequent analysis results. The Venn diagram of bacterial and archaeal OTU data annotated at 97% similarity level was drawn. The similarity and overlap of OTU compositions in the samples were visually displayed, and a rarefaction curve was constructed to check whether the sequencing number was sufficient.

The alpha diversity indices calculated, included Sobs, Chao1, Shannon, and Heip. The Pearson correlation between alpha diversity indices and physicochemical characteristics was performed with SPSS to evaluate the effects of dominant environmental factors on the overall community diversity. The relative abundance of main phyla/classes in lakes was statistically analyzed by Excel. A column chart of the phylum level and a heat map of the genus level were used to analyze dominant and differential species in samples. The beta diversity distance matrix (Bray–Curtis) was calculated using QIIME, and nonmetric multidimensional scaling analysis (NMDS) was performed to cluster the 12 lake samples. Canonical correspondence analysis (CCA) was performed to evaluate the effects of environmental and geographical factors on microbial community structure. The heat map, NMDS, and CCA were performed with the corresponding software package in the Majorbio cloud analysis platform. The spearman correlation test between paired geographical distance and microbial community composition similarity (Bray–Curtis) was used to evaluate the correlation between different lakes.

Sequences generated from the samples used in this study were deposited in the Sequence Read Archive database of NCBI. The accession number for the 12 samples collected from the salt lakes of the Tibetan Plateau was PRJNA884892.

## 3. Results

*3.1. Physical and Chemical Parameters of Lake Sediment*

The 12 lakes studied had different salinity levels, ranging from 0.64PSU to 106.9PSU. According to the salinity level and the range of the most appropriate salt concentration for various halophilic microorganisms [30], the 12 lakes were roughly classified into three groups. Among them, BZC, CXBC, GGC, and PSEC were categorized as low saline lakes (LSL) (salinity < 5PSU); GJLC, DZC, YNPC, and ADEC as moderately saline lakes (MSL) (5 < salinity < 35PSU, equivalent to the salinity of seawater); BGC, DXC, GTC, and YBCK, as high-saline lakes (HSL) (salinity > 35PSU). All lakes are alkaline with pH 8.18–9.24 and an average pH of 8.9. The W was between 12.93 and 58.33. The chlorophyll a content at all stations was low and was the highest at BGC (2.53 µg/g). The DO content fluctuated between 3.73–12.82 mg/L and was the highest in DXC (12.82 mg/L). In the lake with the highest salinity, we measured the maximum dissolved oxygen content, which seemed unreasonable. However, since DXC is backed by snowy mountains, it will be directly recharged by snowmelt water [31]. The inflow of meltwater will increase the fluidity of the lake water and lower the lake temperature, thus increasing dissolved oxygen [32,33]. In addition, we also found that Weelhamby pan in Western Australia, which showed DO values of 14.4 mg/L at a salinity of 112 g/L, which indicated that although salinity has a negative effect on dissolved oxygen it is not a simple linear relationship. For a specific lake, other factors including temperature, pH, and lake mobility may also have impacts on it.

*3.2. Comparison of Alpha Diversity among All Lakes*

A total of 729,046 sequences (49,917–72,119) were retrieved by high-throughput sequencing using the MiSeq sequencing platform, and 599,004 sequences were obtained after flattening. A total of 4972 OTUs were obtained (97% similarity). After random sampling of optimized sequences, the rarefaction curve was constructed based on the number of sequences and their Chao1 index using Mothur software. The results demonstrated that with an increase in the number of sequences, each curve approached asymptotes and all

coverage indices were higher than 98% (Figure S1, Table S1), indicating that the sampling depths were sufficient to capture the overall microbial diversity in all 12 samples.

Alpha diversity indices indicated that the number of OTUs were generally high in LSL and MSL but varied widely in high saline lakes. The highest value appeared in BZC (1814), which had the lowest salinity, whereas the lowest value appeared in DXC (818), which had the highest salinity (Figure 2, Table S1). The changing trend of the Chao1 index was consistent with that of OTU numbers (Figure 2, Table S1). The Shannon and Heip diversity indices showed a decreasing trend in microbial diversity with increasing salinity (Figure 2). In general, the microbial diversity of the LSL was the highest, followed by that of the MSL and HSL (Figure 3, Table S1). Pearson correlation analysis showed that microbial diversity was significantly negatively correlated with increasing salinity (r = −0.661, $p < 0.019$). Accordingly, the relationship between Shannon diversity indices of bacteria and archaea, and salinity was further analyzed. The results indicated a significant negative correlation between the bacterial diversity indices and salinity (r = −0.676, $p < 0.016$). The archaeal diversity indices had no significant correlation with salinity; however, an opposite trend to that of bacteria was observed (r = 0.265, $p < 0.405$) (Figure 3). This suggested that the alpha diversity of bacteria of these lakes was mainly driven by salinity. In contrast, the driving force of salinity on archaea was not very significant. But it showed a general upward trend with the increase in salinity, so salinity may also have a potential impact on it, to some extent.

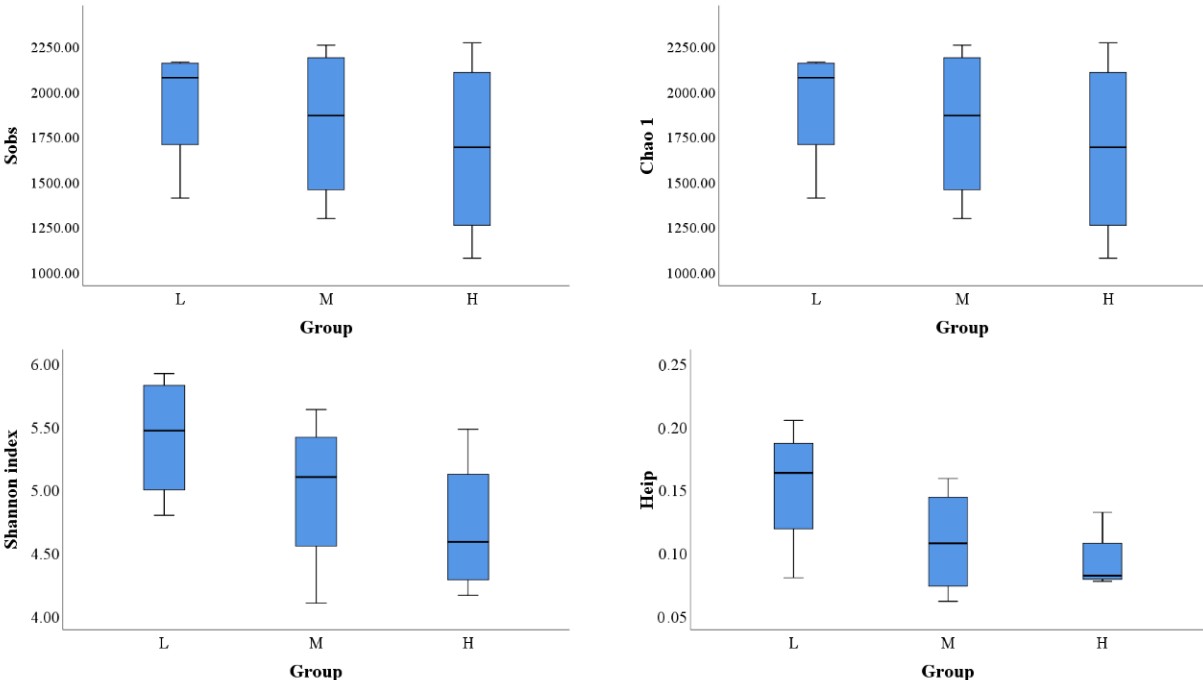

**Figure 2.** Alpha diversity indices of the microbial communities of the sampled lakes on the Tibetan Plateau, China (L: Low saline lakes M: Moderately saline lakes H: High saline lakes).

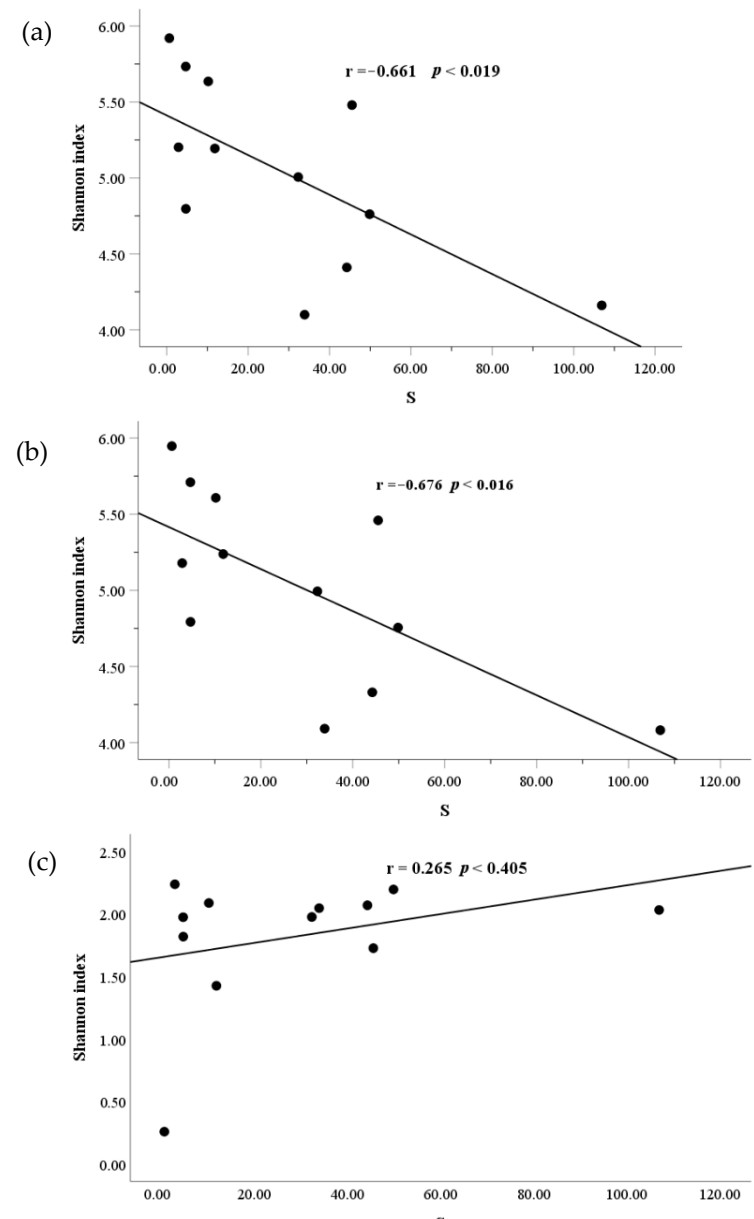

**Figure 3.** Correlation between Shannon diversity indices and salinity, measured for (**a**) All, (**b**) Bacteria, (**c**) Archaea. (S: Salinity).

*3.3. Microbial Community Composition of the 12 Lakes*

Bacteria and archaea represented 97.59% and 2.41% of the total sequences, respectively, and the proportion of bacteria was much higher than that of archaea. The Venn diagram showed that LSL and MSL shared 909 OTUs, MSL and HSL shared 330 OTUs, LSL and HSL shared 283 OTUs, which was the lowest number, and 1522 OTUs were shared by all three groups (Figure S2).

*Proteobacteria* (28.89%), *Bacteroidota* (15.67%), *Firmicutes* (15.64%), *Actinobacteriota* (12.86%), and *Chloroflexi* (5.58%) were the most abundant bacterial phyla detected in all lake samples (Figure S3). The relative abundance of *Firmicutes*, *Desulfobacterota*, and *Halanaerobiaeota* in the three lake groups showed an increasing trend with salinity. In contrast, *Chloroflexi*, *Gemmatimonadota*, and *Acidobacteriota* exhibited opposite trends, their proportions decreased significantly with increasing salinity (Figures 4 and 5). Archaea mainly included *Crenarchaeota* (70.83%), *Halobacterota* (16.25%), and *Nanoarchaeota* (12.29%) at the phylum level (Figure S3). *Crenarchaeota* was the dominant phylum in LSL and MSL, account-

ing for 85% and 91.55%, respectively. *Halobacterota*, which was almost undetected in MSL, increased significantly in HSL, accounting for 39.38%, whereas *Crenarchaeota* decreased to 35.63%. The relative abundance of *Nanoarchaeota* increased with increasing salinity (Figure 4). The Kruskal–Wallis test was applied to carry out a comparative analysis, and the results indicated that *Gemmatimonadota*, *Acidobacteriota*, and *Halanaerobiaeota* exhibited significant differences among the three lake groups (Figure S4). Compared to the high abundance of *Gemmatimonadota* and *Acidobacteriota* in LSL, that of *Halanaerobiaeota* in HSL was higher (Figure 5).

The 40 most abundant genera of bacteria and archaea in all samples were analyzed using a hierarchical clustering heat map, and the results are shown in Figure 6. The main genera were divided into three distinct groups. Cluster I mainly consisted of non-halophilic species, including *Longimicrobiaceae* (family), *Truepera*, *Aquiflexum*, *and Azoarcus*, which were mainly distributed in LSL and some MSL. Cluster II was composed of some bacteria that were less tolerant to salinity, such as *Planococcaceae*, *Rhodoferax*, *Fusibacter*, *Anaerolineaceae*, *Arthrobacter*, *and Proteiniclasticum*. Some moderately halophilic and halotolerant bacteria were the main members of Cluster III and were mainly distributed in HSL. For example, *Marinobacter*, *Anoxynatronum*, *Thioalkalivibrio*, *Halomonas*, *and Balneolaceae*, *Peptostreptococcaceae* (family).

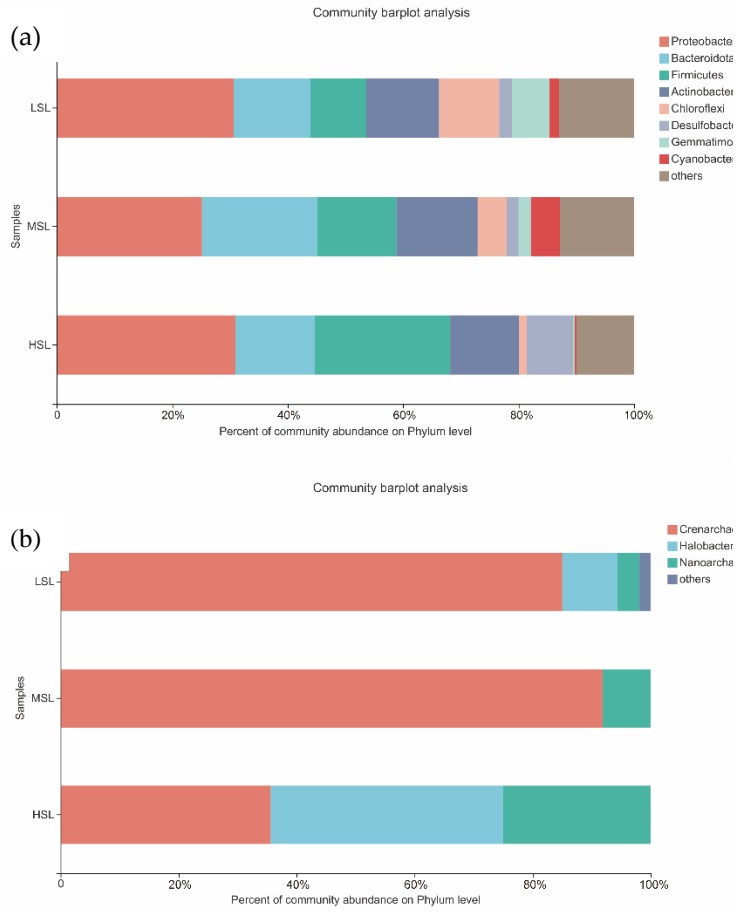

**Figure 4.** Composition of microbes of the sampled lakes on the Tibetan Plateau, China at phylum level: (**a**) Bacteria, (**b**) Archaea.

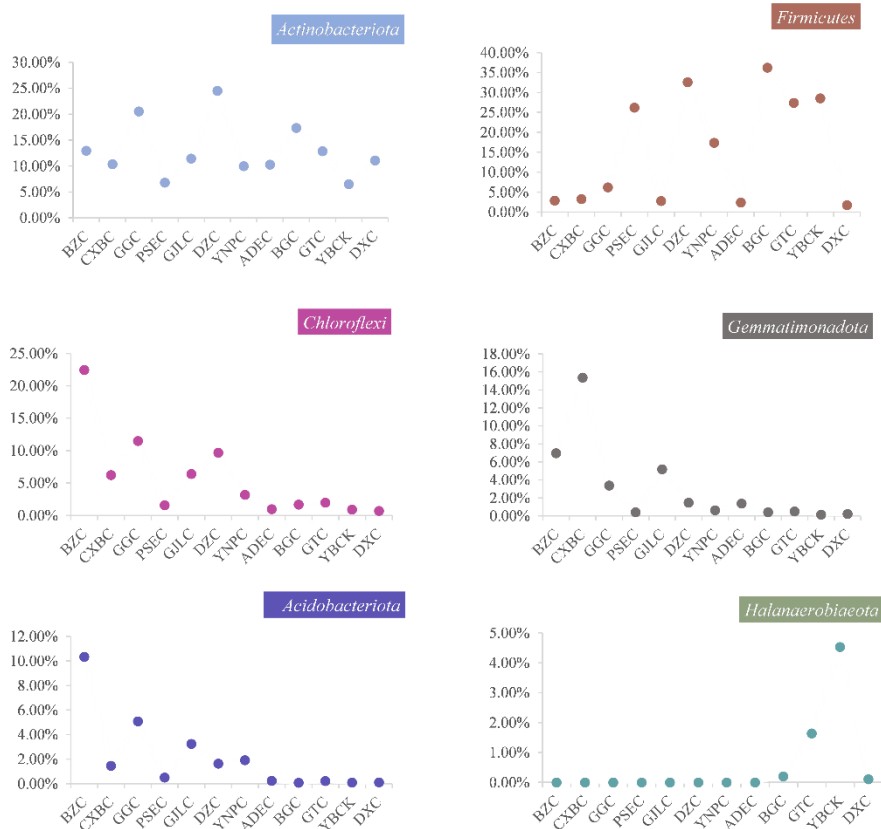

**Figure 5.** The relative abundance of the major phyla/classes along a salinity gradient in the 12 lakes on the Tibetan Plateau, China.

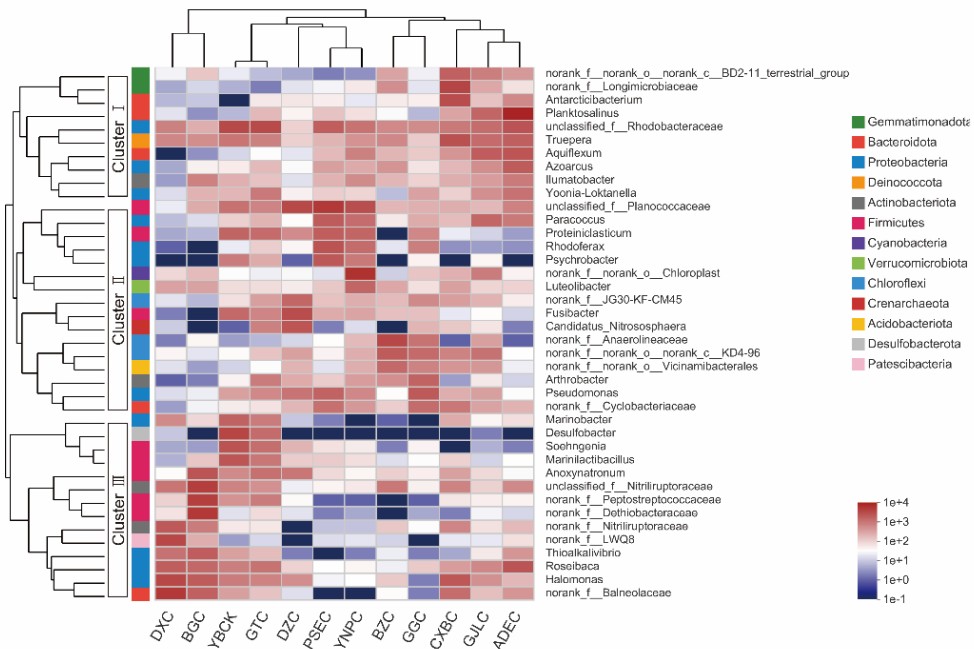

**Figure 6.** The heatmap shows the relative abundance of the top 40 abundant genera with color legend.

GTC and YBCK in the HSL, apart from being located close to each other, also had similar salinity, and microbial communities. The abundance of *Desulfobacter*, *Soehngenia*, and *Marinilactibacillus* in the two lakes were significantly higher than that in the other lakes. In addition, the abundance of *Desulfobacter* in the phylum *Desulfobacterota*, which was

hardly detected in the other lakes, was also extremely high. The abundance of *Chloroplastno*, *Anaerolineaceae*, *Planktosalinus*, *Aquiflexum*, *Longimicrobiaceae*, and *Antarcticibacterium* in low and moderately saline lakes was higher than that in high saline lakes. The abundance of *Anoxynatronum*, *Nitriliruptoraceae*, *Peptostreptococcaceae*, *Dethiobacteraceae*, *Thioalkalivibrio*, *Nitriliruptoraceae*, *Halomonas* and *Balneolaceae* was high in the high saline lakes; however, it decreased significantly in the low and moderately saline lakes. This phenomenon also indicates a shift in the dominant microbial taxa from low and moderately saline lakes to high saline lakes. Although the salinity of BGC was similar to that of GTC and YBCK, in terms of microbial composition, GTC and YBCK were more similar. The BGC had different microbial composition from that of the two lakes, but similar to that of DXC (salinity 106PSU) with a large difference in salinity. This suggests that although salinity has an important influence on microbial community composition at the overall scale, other local factors may also play a potential role in the distribution of microbial communities in the lake to some extent.

### 3.4. Effect of Physicochemical Factors on Microbial Composition of Lakes

NMDS of the Bray –Curtis distance was performed to visualize the relationship between the microbial communities in the 12 samples. When the stress is less than 0.1, it can be considered a good order. The NMDS results showed that in terms of microbial taxa composition on OTU level, the HSL samples were more concentrated than those of low and moderately saline lakes, and the MSL samples were the most dispersed. In addition, the LSL and HSL could be significantly separated, but MSL had overlapping areas with both (Figure 7).

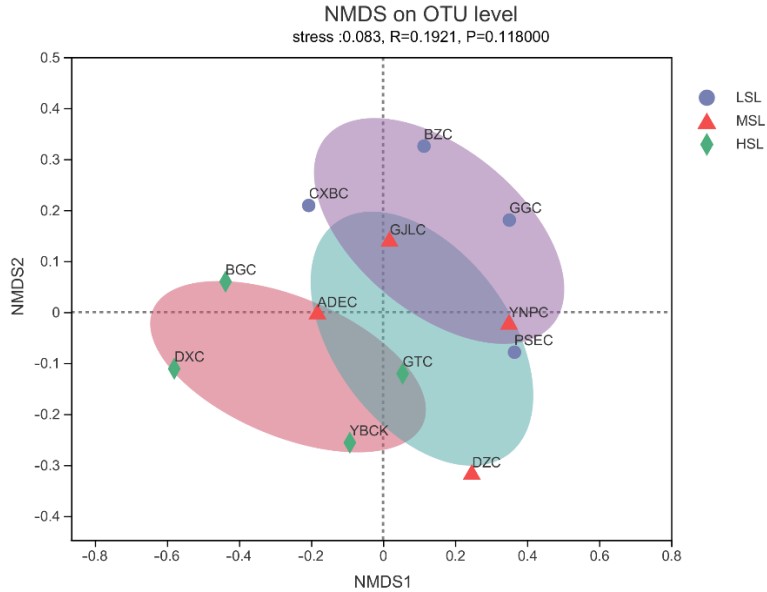

**Figure 7.** Non-metric multidimensional scaling (NMDS) of microbial community composition in the sampled lakes on the Tibetan Plateau, China.

CCA demonstrated that the composition of the lake microbial community was related to salinity, DO content, and chlorophyll content (Figure 8). Among these factors, salinity had the highest explanatory power (18.37%, r = 0.779, *p* < 0.001). The microbial community compositions of low and moderately saline lakes were negatively correlated with salinity, and geographical factors, including relative distance to the glacier, and had important effects on it. In addition, we also found that some species were significantly correlated with some specific factors. For example, *Antarcticibacterium*, *Planktosalinus*, and *Aquiflexum*, with high abundance in CXBC, GJLC, and ADEC, were all significantly negatively correlated with OM (Figure S5).

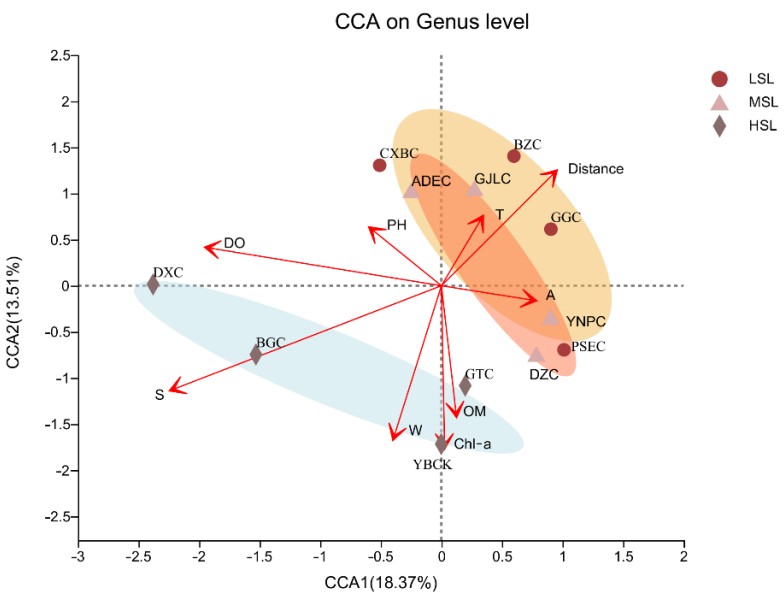

**Figure 8.** Canonical correspondence analysis (CCA) of microbial community composition and physicochemical characteristics of the sampled lakes on the Tibetan Plateau, China.

However, there was a significant positive correlation between microbial community composition and salinity in HSL. DO content was another major environmental factor that had a similar effect to salinity. Chl-*a* content was also an important factor. In addition, there was no correlation (r = 0.080, *p* < 0.523) between the Bray–Curtis dissimilarity in microbial composition and the paired geographic distance between these lakes (Figure 9). Therefore, geographical distance had no significant effect on the microbial composition of the samples.

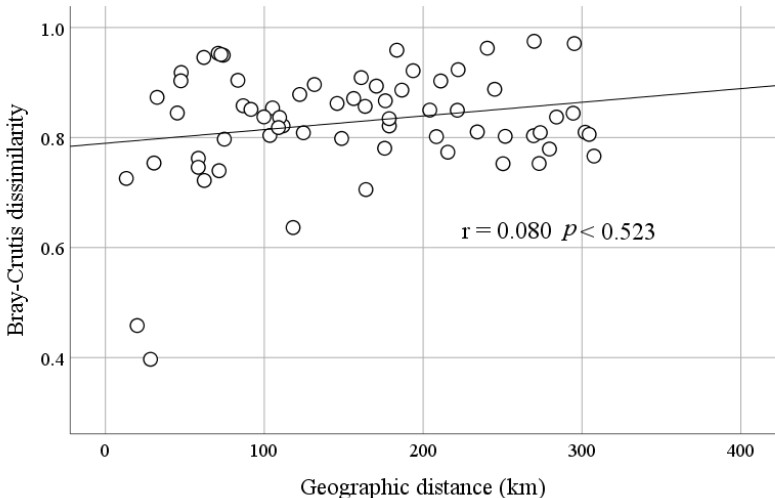

**Figure 9.** Spearman correlation between Bray–Curtis dissimilarities of microbial communities and geographic distance among the sampled lakes on the Tibetan Plateau, China.

## 4. Discussion

### 4.1. Distribution of Microbial Community Composition along a Salinity Gradient

In this study, we found that *Proteobacteria* and *Bacteroidota* were the dominant bacterial phyla in lakes with different salinity (low, moderately, and high), the result was the same as previous research. Ben et al. studied the microbial community structure of the bottom sediments of the Jereed Great Salt Lake in southern Tunisia during the rainy season. The results showed that the bacterial community in the sediments was dominated by *Proteobacteria*, followed by *Firmicutes* and *Bacteroidota* [34]. In the study of the bacterial

community structure of Sanggendalai, Dagenao and Zhagestainao in the Hunshandake area of Inner Mongolia, it was found that the bacterial community of Sanggendalai saline-alkali lake was mainly composed of *Proteobacteria*, Dagenao salt lake was mainly composed of *Cyanophyta*, Zhagestainao salt lake was mainly composed of *Chloroflexi* [35]. Several studies have found that the microbial taxa of salt lakes were dominated by the *Proteobacteria*, *Firmicutes*, *Bacteroidota* and *Acidobacteriota* [36,37] However, differences in the physicochemical composition of individual salt lakes causes differences in the proportion of dominant taxa. The species and genetic diversity of *Proteobacteria* are extremely rich, which determines that the group covers a very broad range of metabolic types. They are abundant not only in oceans and high-salinity waters, but also in freshwater habitats [38–40]. *Bacteroidota* have been found to be ubiquitous on the estuarine gradient [41], and in some cases they are also the dominant group in higher salinity environments [42]. In addition, many of the salt-tolerant bacteria currently capable of growing in widespread levels of salinity are found to belong to *Proteobacteria* and *Bacteroidota* [43,44]. Salt-tolerant bacteria can rely on several common strategies to grow and survive in a large salinity range, resulting in the wide distribution of both in low, moderate and high salinity lakes. Although the proportion of *Firmicutes* in low and moderately saline lakes was lower than that of the former two, it was higher than that of Bacteroidota, in high saline lakes accounting for 23.48%. Previous studies have found that this phylum contains a large number of halophilic bacteria, such as *Halobacillus* spp. [45]. There are two main mechanisms by which microorganisms adapt to saline environments. Most halotolerant microorganisms and moderate halophiles respond to the increase in salinity by accumulating compatible solutes: small molecule organic compounds (e.g., amino acids and carbohydrates). The cell metabolism is undisturbed even at very high concentrations. In addition, osmotic balance can be achieved by accumulating KCl as the only, or the main, osmotic solute. The salt tolerance mechanism of microorganisms in Proteobacteria is mainly the former, while the latter is applicable to *Bacteroidota*; however, both are available for *Firmicutes*, which may also contribute to the increased *Firmicutes* proportion in progressively higher salinity gradients [46,47]. The numbers of *Chloroflexi*, *Gemmatimonadota*, and *Acidobacteriota* in LSL were higher than those in the other lakes. Some known taxa of *Gemmatimonadota* are mainly isolated from soil, plant rhizospheres, and freshwater lakes, and the culture environment is aerobic and low salt [48]. *Vicinamibacteraceae*, a member of *Acidobacteriota*, is a gram-negative bacterium that grows in an aerobic, neutrophilic environment but has a very low tolerance to NaCl [49], resulting in its highest abundance in BZC with the lowest salinity and very low abundance in other lakes.

Clusters I and II were dominated by some freshwater non-saline and micro-saline bacteria, and some also showed obvious anaerobic or aerobic characteristics. *Aquiflexum*, mainly isolated from lake water, is a gram-negative facultative anaerobe that grows at pH 6.5–8.5 and 0–2.0% (*w/v*) NaCl (optimum 0–0.5%) [50]. *Candidalus_nitrososphaera* is a type of ammonia-oxidizing archaean that is non-saline or micro-saline and can grow in a 30 PSU environment [51]. Previous studies on 16 alpine lakes in Tibet have found that 70 % of freshwater microbial groups are limited at a salinity of ≤2 PSU, laboratory cultures have found that they can be tolerated to a degree of about 10PSU [39]. The main representative groups in the low salinity salt lake group (<30PSU) in the Badain Jaran Desert are also freshwater and slightly halophilic groups, including *Limnohabitans*, *Flavobacterium*, *Burkholderia*, *Pseudomonas*, and *Arthrospira* [52]. But their tolerance to salinity is very limited. With the increase in salinity, the abundance of these genera gradually decreased and was gradually replaced by groups with better salt tolerance. The moderately halophilic or halotolerant bacteria in Cluster III mainly belong to *Firmicutes* and *Proteobacteria*, and were concentrated in HSL, and these taxa were significantly positively correlated with salinity (Figure S5). Typical of these was *Halomonas*. *Halomonas* belongs to the *Halomonadaceae* family of the *Gammaproteobacteria*. Many bacterium species have been isolated from different saline–alkali environments. They are moderately halophilic, aerobic, and can tolerate up to 20% salinity [53]. Yang et al. isolated 13 *Halomonas* strains with different

morphological characteristics from five salt lakes in Tibet with salinities ranging from 40-110PSU, indicating a wide salt tolerance range of *Halomonas* [54]. Not only that, *Halomonas* was distributed in Darban Salt Lake in Xinjiang, LopNor Salt Lake and Dagu Salt Wells in Sichuan, and all of them were the dominant flora [55]. *Soehngenia*, and *Marinilactibacillus* of *Firmicutes* are the main anaerobic salt-tolerant bacteria [56,57]. In addition, *Halanaerobiaeota* is mainly composed of anaerobic fermentation bacteria and has a certain tolerance to salinity [58]. *Thioalkalivibrio* is a sulfur-oxidizing bacterium. With an increase in lake water salinity, the osmotic pressure that microorganisms need to resist also increases. The composition of the sulfur-oxidizing bacterial community also changed from freshwater taxa to salt-tolerant and halophilic taxa. Among them, *Thioalkalivibrio* is the dominant taxon in high saline–alkali lakes [59]. Several researchers have found a predominance of halophilic bacteria in ephemeral saline lakes in the Monegros Desert (northeastern Spain) with highly variable salinity, although these ephemeral lakes sometimes reach near saturation [60]. The study also found that soda lakes with salinities of 35–250PSU have functional and diverse halophilic alkaline microbial communities, mainly responsible for the cycling of chemical elements such as carbon, nitrogen and sulfur [61]. Elevated levels of sodium ions in high salinity environments have toxic effects on biomolecules such as nucleic acids and proteins in cells, so salophilic organisms must adopt preventive mechanisms to avoid the entry of excess sodium ions into cells, which most salophilic bacteria can resist through the "salt precipitation" strategy. In addition, under conditions of high salinity and low nutrients, microorganisms may adjust their survival mode by activating stress responses and detoxification and repair mechanisms so that they can better adapt to high salt environments [62]. The energy required to live in a hypersaline environment is expensive and the limit of salt concentration to support different heterotrophic processes may be determined by bioenergetic limitations [63].

The adaptation mechanisms of microorganisms to different salinities is related to osmotic stress. In addition, salinity seems to be associated with differences in key metabolic capacities of bacteria, including large differences in the relative abundance of genes involved in respiration, glycolysis, biosynthesis of aminophenols, and osmoregulation [42]. It has been proposed that the effect of salinity on microbial function may potentially be regulated by microbial community composition, which is attributed to the preference of microorganisms for salinity in phylogeny [64]. However, the functional differences between these communities and the reasons behind these differences have not been fully explored. Reduction of biodiversity at high salinity may further contribute to the reduction in microbial function and activity. Previous studies on wetland sediments found that high salinity exposure reduced phosphatase and N-acetylglucosaminidase activity by 20% [65]. The potential decrease in microbial activity and metabolic function further emphasizes the importance of studying which specific microbial communities can survive in saline environments with inhibitory diversity [66]. Changes in microbial communities may also further alter specific microbial metabolic pathways in response to salinity changes. For example, it has been recorded that the addition of artificial seawater in the laboratory changed the mineralization pathway, from methane production to iron reduction, and finally sulfate reduction was detected in sediment samples as dominant [67]. A recent study of salt pans along the coast of South Africa, based on the amplification of the dissimilatory sulfite reductase gene, showed that the activity of *Desulfobacteraceae* was very high at salinities up to 300–400PSU [68]. In the sediments of high salinity lakes, thiosulfate and elemental sulfur are better electron acceptors than sulfate [69]. Studies have shown that some genera contained in *Dethiobacteraceae* can use sulfite or thiosulfate as an electron acceptor [70], which may also be one of the important reasons why *Dethiobacteraceae* can exist in high-salinity lakes in this study. With the development of metagenomics, significant changes in microbial gene abundance and expression patterns of genes encoding catabolic pathways, osmoregulation, and metal transporters have been observed along river-crossing to ocean salinity gradients, showing key differences in gene abundance for bacterial respiration, glycolysis, osmoregulation, and metal transport at different salinities [71]. These support-

ing microbial functional profiles may be significantly affected by changes in community composition caused by salinity changes. Therefore, future work should directly evaluate microbial functional profiles using high-resolution metagenomics and metatranscriptomics to determine specific microbial processes in response to salinity changes.

Archaea are widely distributed and occupy various niches. Organic and inorganic electron donors and receptors can be used to evolve various energy metabolism pathways. However, the archaeal ecology in alpine lake sediments has received less attention. This study demonstrated that archaea only accounted for 2.41% of the total sequence, with a low proportion. *Crenarchaeota* were detected in all lake sediments and were ubiquitous in the low and moderately saline lakes. Most of these are anaerobic heterotrophic organisms that mainly use proteins and sugars for various physiological and biochemical reactions, and a few are sulfur (oxidation and reduction) cyclic chemical autotrophic organisms [72]. A large number of anaerobic methanotrophic taxa in *Halobacterota*, *Methanophagaceae*, belong to the ANME-1 group, which can survive in a wide temperature range and high salt environment [73,74].

### 4.2. Effects of Environmental and Spatial Factors on Microbial Communities in these Lakes

Among several environmental factors, salinity had the most significant effect on microbial diversity and the microbial community of surface sediments. Statistical analysis showed that salinity was significantly negatively correlated with microbial diversity indices, indicating that salinity greatly limited the microbial diversity. CCA results also showed that salinity contributed the most to the microbial community composition. Microbial activity in salt lakes is mainly controlled by salinity due to energy limitations, and therefore salinity should be the main factor influencing microbial diversity and community composition, and our results were also consistent with previous results [36,39,75–77].

The significant effects of geographic distance have been explored in lakes at various spatial scales in different regions [13,78,79]. However, in this study, there was no significant correlation between geographical distance and microbial community distribution in the present study (r = 0.080, $p < 0.523$). This inconsistency may be due to the difference in distances between previously studied lakes and those studied in the present study (13–307 km vs. 4–1670 km) [13]. Previous studies also showed that geographic distance was negatively correlated with similarity in bacterial communities between the Antarctic, the Arctic, and the Tibetan Plateau, but all three spanned large continental scales [80]. The influence of spatial factors on microbial distribution patterns may be scale dependent. Within a small range (<500 km), contemporary environmental factors such as pH and DO content are often the main factors affecting the distribution of microbial communities [36]; at tens of thousands of kilometers, the geographical isolation effect generated by spatial factors plays a more apparent role in the change of microbial community [81]. Therefore, it was reasonable to observe the effect of salinity on the microbial community structure at small spatial scales, where the distance effect was not evident.

Although salinity was the most important environmental impact factor, the hierarchical clustering analysis results of species composition at the genus level also showed cross-clustering phenomena between lakes in moderately and low saline lakes. In terms of species composition, the moderately and low saline lakes were more similar, but the high saline lakes showed significant differences. CCA results also demonstrated that the environmental factors affecting the microbial community composition in moderately and low saline lakes were similar, and significantly different from those in high saline lakes. There are two possible reasons for this result. First, the salinity range of the samples collected in this study was narrow, and the salinity of most lakes was below 50PSU, resulting in clustering analysis results that were not as obvious as those of other studies with large salinity range [11,52]. Another reason is that, although salinity played an important role in the construction of microbial communities, it cannot completely eliminate the influence of other local characteristics in driving the distribution of specific microbial categories, that is, it was affected by multiple environmental factors. This was consistent

with the report of halophilic bacteria in Antarctica and Cariboo Plateau, British Columbia, Canada [82]. Previous studies on five lakes in the central and southern Tibetan Plateau showed that the microbial community structure was affected by the combined effects of salinity, pH, chl-*a* content and lake hydraulic retention time [36]. Due to the harsh natural conditions of salt lakes, only some low algae and some special animals can barely adapt and primary productivity is dominated by phytoplankton. Generally, lakes with salinity over 20g/L have no more grass present [83]. In this study, the proportion of photoautotrophic *cyanobacteria* was also low, accounting for only 2.07%. Although the sampled sites were located in the coastal area with strong light. But the lack of nutrients in the lake itself may limit the survival of such microbes [84]. The Chl-*a* content in the sediments we measured was also relatively low, mostly concentrated between 0.02–0.67ug/g, and in a few lakes such as PSEC were higher than 1ug/g, indicating that the concentration of benthic algae in the lake was low. *Proteiniclasticun*, *Planococcaceae*, and *Marinilactibacillus* were significantly positively correlated with Chl-*a* content (Figure S5), and they were the most abundant in PSEC and YNPC with high Chl-*a* content, indicating that benthic algae may also affect some groups to some extent. Algae extinction in waters is closely related to nitrogen deposition, which also affects the metabolic activity of microbes of the surface sediment [85]. In the special ecosystem of salt lakes on the Tibetan Plateau, benthic animals mainly include *Daphnia tibetana*, *Artemia* etc., but these species also have a range of adaptation to salinity. *Artemia* mainly grows in water bodies with salinities of 50–200 g/L, but nearly 40.74% of the salt lakes are still without distribution, the survey showed that they are distributed in the high salinity lakes, YBCK etc. [83]. Halophilic bacteria are very diverse in their ability to decompose organic substrates. So far, a rarely developed substrate is chitin, which is produced in large quantities in high-salinity habitats due to high biomass production by *Artemia* [63,86]. In the current study of microbial communities from highly saline lakes in the Kulunda grassland (Aratai, Russia), active moderately saline-tolerant chitin degrading microbial communities were found in the high saline lakes, and many alkalophilic chitin-degrading bacteria were isolated from the soda lake sediments and surrounding soils [86]. Therefore, it was speculated that these important groups in high-salinity lakes may also use chitin as an organic substrate for metabolic activities, but this also requires further study. In addition, we also found that *Aquiflexum*, *Planktosalinus*, and *Antarcticibacterium* (all belonging to *Bacteroidota*) were significantly negatively correlated with OM, and were mainly distributed in stations such as GJLC, CXBC, and ADEC with extremely low organic matter content, which may be attributed to their ability to degrade complex organic matter [87]. Previous study has found considerable diversity of *Bacteroidota* in the nutrient-poor eastern Mediterranean [87]. Lakes YBCK and GTC are close to glaciers, and both are fed by glaciers. Previous studies on Ranwu Lake have shown that the seasonal melting of glaciers in summer increases meltwater in a short period of time, which reduces the electrical conductivity of the lake and thus benefits the survival of freshwater microbial communities. At the same time, the microbial communities remaining in the meltwater will also enter the lakes [33], which may be one of the important reasons for the clustering of the two lakes with LSL. The study of prokaryotic diversity in the waters of the Atalazo Salt Lake in the Atacama Desert in northern Chile found that the microbial community structure also changed with increasing latitude [88]. The study on sediments at different depths in Qinghai Lake also showed that the proportion of *Proteobacteria* decreased with the increase in depth, while the proportion of *Firmicutes* increased [14]. Although there was no detailed specific study in this paper, it also provided new ideas and directions for future research.

Geographically distant lakes with similar salinity had similar microbial community compositions, such as BZC and GGC; however, some geographically close lakes had a large difference in salinity, which led to different microbial community compositions between them, such as PSEC and ADEC. Therefore, this study also supported Beijerinck's hypothesis that "everything is omnipresent but the environment is selective" [89,90], which implies

that environmental filtration plays an extremely important role in the formation of microbial communities [91].

## 5. Conclusions

The Tibetan Plateau is a remote high-altitude area, far from human habitation, and the lakes in the region are mostly alkaline and span different salinity levels (0.64–106.9 g/L). The level of bacterial diversity in the study area was greatly limited by salinity, whereas archaeal diversity showed an increasing trend. *Proteobacteria* and *Bacteroidota* were the main bacteria. The microbial community composition of MSL was similar to that of LSL. There were a large number of tolerant and halophilic organisms in HSL, including some dominant groups of *Firmicutes* and *Halanaerobiaeota*. The proportion of *halobacterota* in archaeans increased significantly. Statistical analysis demonstrated that it was not spatial factors but physicochemical factors that influenced the composition of microbial communities in lakes on the Tibetan Plateau. Salinity was the most important factor restricting alpha diversity, and it was also the main driving force for microbial community aggregation. In contrast, geographical factors had no significant effect on the microbial community composition of the lakes on the Tibetan Plateau. However, this study also provided basic reference information for further improvement of the sampling scale in the future, and for studying the characteristics of microbial community results in relatively less disturbed ecosystems. Therefore, these observations provided baseline information for predicting the response to future environmental changes in the region.

**Supplementary Materials:** The following supporting information can be downloaded at: https://www.mdpi.com/article/10.3390/w14244043/s1, Figure S1: Rarefaction curves of sediment microbes in the sampled sites on the Tibetan Plateau, China, Figure S2: Venn diagram showing the number and proportion of shared and unique OTUs of the sampled lakes on the Tibetan Plateau, China, Figure S3: Composition of microbes of the sampled lakes on the Tibetan Plateau, China at phylum level, Figure S4: The Kruskal–Wallis H test of low, moderate and high saline lakes on the Tibetan Plateau, China at phylum level, Figure S5: Spearman correlation of sediment microbial composition and environmental factors by heatmap, Table S1: The alpha diversity indices of the 12 sampled lakes on the Tibetan Plateau, China.

**Author Contributions:** Methodology, data analysis, and writing—original draft preparation, Y.H.; methodology, L.H.; data acquisition and process, Z.W. and T.L.; project administration, S.S.; writing—review and editing, X.L. All authors have read and agreed to the published version of the manuscript.

**Funding:** This research was funded by the Fundamental Research Funds for the Central Universities, grant number 201964024 and the Science and Technology Project of the Tibet Autonomous Region, grant numbers XZ201703-GB-04 and XZ202102YD0022C.

**Data Availability Statement:** The sequence data presented in this study can be accessed in the SRA database of the NCBI with accession number PRJNA884892.

**Acknowledgments:** We are grateful for the assistance provided by Long Hongan, Wang Qi, Zhu Boshan, and Wang Pengfei in field sampling in Tibet, China.

**Conflicts of Interest:** The authors declare that they have no known competing financial interests or personal relationships that could have appeared to influence the work reported in this study.

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
