# Peer review of "Salinity Shapes the Microbial Communities in Surface Sediments of Salt Lakes on the Tibetan Plateau, China"

_water, doi:10.3390/w14244043_

Round 1
Reviewer 1 Report (Previous Reviewer 1)
It’s an interesting study. But there are some minor revisions need to be done.
Some specific suggestion:
1, Figure 1: The scale is incorrect.
2, The “‰” were out of date in international regulation. Salinity is no dimensionless.
3, L128-132: Please correct the subscript in the chemical formula.
4, Line137: All the methods mentioned above are not in the standard of 12763.1-2007.
5, Figure 2: The first letter of “Sobs” and “Heip” should be capitalized. “Shannon” Should be used as “Shannon index”. The numbers in Y-coordinate are too small to see clearly.
6, Figure 3: (a) is missing in the first figure. “Shannon” Should use as “Shannon index”.
7, Figure 9: The “Km” should use in lowercase.
Author Response
Please see the attachment.

Reviewer 2 Report (Previous Reviewer 2)
Thank you for your attention to my comments. But your English still needs to be corrected. Please check the spelling of the taxa.
Author Response
Please see the attachment.

Reviewer 3 Report (New Reviewer)
The study on microbial communities in sediment of salt lakes in the Tibetan Plateau provides interesting and new information on microbial communities in these extreme environments. Conclusions on the study are that Proteobacteria and Bacteroidota were the major bacterial groups and that Crenarchaeota were the dominant archaea in the low-moderately saline lakes, whereas Firmicutes and Halobacterota were more common in the high saline lakes.
The molecular work and the bioinformatic and statistical treatments are performed very well and are adequately presented. However, I have two main concerns with this study. Firstly, all of the biological and bioinformatic conclusions are based upon few samples collected near the shore at maximum 0.5 m depth in each lake. How can the authors know if the results are representative for the microbial communities at shallow sediment depths in each of the lakes? Do the results just represent microbial communities at the surface sediment at one location in each of the lakes? I have no information on biology of the lakes, but I assume there might be vegetation or some animal life that can impact the microbial communities at larger depths and at different locations with other types of sediment.
Secondly, I miss some presentation of biological functions of the different microbial groups. For example, why are Proteobacteria and Bacteoidota dominant at low salinity? Do microbiological or metabolic functions other than tolerance to salinity affect the community compositions? Could intense sunlight at low water depths be a shaping factor? What about algal grow, e.g., om sand grains, etc.
I strongly recommend that the authors provide information on these two issues.
Author Response
Please see the attachment.

Reviewer 4 Report (New Reviewer)
The manuscript entitled "Salinity Shapes the Microbial Communities in Surface Sediments of Salt Lakes in the Tibetan Plateau, China" refers to interesting aspects concerning the diversity of microbial communities in the surface sediments from 12 lakes which differ in salinity level and their response to changing environmental conditions. However, the minor revision should be done, especially to improve the section “Results” and the section “Discussion”.
Detailed main comments
1. What do you mean about “water content (W)”? Please clarify this expression.
2. Why did you use “The Pearson correlation test”? Were the date normally distributed?
3. Why do you cite references and discuss the results in the section “Results”? (For example: lines 179, 188-195, 226-229, 261-263, 265-266, 272-273, 278-280,
Although, you discussed the results in the section “Discussion”. Please clarify the content of both sections.
4. Please explain the codes used in Figures 4, 5, 6, 7, 8.
5. Do you really need to cite the Figures in the section “Discussion”?
Author Response
Please see the attachment.

This manuscript is a resubmission of an earlier submission. The following is a list of the peer review reports and author responses from that submission.
Round 1
Reviewer 1 Report
It’s an interesting study and the manuscript is good in arrangement. However, I think the quality maybe improve after some minor revision.
Some specific suggestion:
1, Line15-17: It may be better if the sentence organized as “the microbial community composition in the surface sediments from 12 lakes with different salinity in Tibetan were analyzed through the Illumina high-throughput sequencing platform.”
2, Line72-73: There are no specific investigation and analyze about glacial meltwater, geographic distance, latitude and longitude in the paper. Then, it is inappropriate to answer the questions (2) and (3).
3, Figure 1: I suggest that “China” be marked in the territory of China. Note the position of glacier. There is a superfluous triangle shape near BGC.
4, Please add information about sampling, such as the number of parallel samples in each lake and the number of DNA extraction for each sample.
5, Line 107 Is there mistake in(550A, ,,, USA)?
6, line 104 and 110: “water content” or “moisture contents”, Please use a consistent name.
7, Line111: 12763.1-2007 is general rule for this standard. I suggest state the specific standard for related item.
8, Line 145: “them” should be “then”.
9, Line217-219: The appearing order of Figure 5 and Figure 6 in the text doesn’t follow the order of numbers. And I think Figure 6 is unnecessary.
10, Please check the salinity should be written as “PSU” or “‰”.
11, Figure 4: The name of x-coordinate is “percent of …”, but labelled in decimals.
12, Figure 9: Where is (a) or (b) figure?
13, The “1” in “Chao 1” was missing in several part in the manuscript and SI.
Reviewer 2 Report
The study of diversity in high-mountain lakes with different salinity is a very interesting and up-to-date direction in microbiology. The authors studied a large number of lakes (12) with different salinity and discussed the effect of this factor on the composition and diversity of microbial communities. And data of diversity in the salinity lakes are of very interest. Both common OTUs (operational taxonomic units) for all lakes and individual OTUs were identified, it enables to assess the metabolic potential of communities in the sediments of the studied lakes. Undoubtedly, such research deserves to be published. I need to note some points of the proposed article that require revision. The authors should check spelling of the taxon’s carefully. It would be useful to indicate what name this or that taxon had previously, so that it would be easier to compare to the earlier published data. Throughout the text, follow, please, the same naming system; be guided by SILVA 138.
In the introduction. The authors did not show the importance of studying these particular lakes, their difference from other lakes in this region and other salt lakes of the world. It is necessary to expand this section and show the composition of microbial communities in salt lakes under another climate and in other regions of the world. The authors need to demonstrate the uniqueness of the studied lake ecosystems and. The authors must demonstrate what is interesting about these lakes and their difference with similar ecosystems.
Research areas
77- Which type of sampler was used for sampling of the sediments? Was it benthic corer, what type? Do the lakes differ in depth of the water column? Where were taken the sediment samples? In the coastal zone?
125 - SLIVA 138 correct, please, to the right variant SILVA
149 - Results
189- 193 Indicators of archaea diversity had no significant correlation with salinity; however, there was an inverse trend compared to bacteria (r = 0.265, p < 0.405) (Fig. 3). This indicates that the alpha diversity of these lakes is mainly determined by salinity, which constitutes not only the species diversity of bacteria and archaea, but also their distribution in a certain spatial range. It is not quite clear - there are no reliable correlation coefficients between archaea and salinity, but there is an effect on species diversity. Explain, please.
240-249 Geography cannot influence the diversity; you have noted other conditions in this lake, which determine the difference in the composition of microbial communities.
Fig. 5 It would be better to use a single scale in the Y-axis to better perception of the difference in the contributions of individual taxa on the graphs.
264-265 Samples were more concentrated – It means the salinity or number of taxa?
278 Therefore, the geographical isolation had no significant effect on the microbial composition of the samples. Are they really isolated? Is there any intake of glacial water after all?
320 The currently known Anaerolineaceae are all thermophilic or thermophilic- What is this?
Discussion
The discussion largely repeats the results of the study. We would like to understand what type of metabolism is characteristic of microbial communities in lakes with different salinities. Despite the proximity of the lakes, they differed significantly in physicochemical parameters that affect the metabolism and diversity of microorganisms. It is likely that the composition of communities may change depending on the season of study, air and water column temperatures, as well as on the trophic status of the water body and the concentration of biogenic elements. In this regard, you should compare the results of your studies of microbial communities with data from studies of salt lakes in other regions of the world. It is fundamentally important to make your research interesting for others and to show the features of the lakes studied by the authors.
Reviewer 3 Report
Dear authors!
You have done a lot of work, but unfortunately, I see a fundamental problem that does not allow me to recommend this article for publication.
You analyzed water parameters and you analyzed the composition of microbial communities in surface sediments. The problem is that water column and sediments are completely different econiches, especially in salt lakes. It is known that physicochemical gradients may appear at the transition from water to sediment. Salinity and pH can change; also as a rule, the dissolved oxygen, temperature and organic matter changes dramatically. In addition, such parameters as the content of sulfates and/or methane are of paramount importance. Therefore, the water column and sediment communities differ significantly. The chlorophyll in the water column originates from plankton and usually is unrelated to sediment communities. As a summary: your critical mistake is that you analyze the diversity of communities in one ecological niche (sediments) depending on factors from another ecological niche (water column). I don't see any scientific reason for this.
If you have a desire to somehow rethink the data obtained, I would recommend, firstly, to take into account the comment above, and secondly, to describe the materials and methods in much more detail. The interpretation of your results can be influenced by the total depth of the lakes, the depth from which water samples were taken for analysis, the depth at which sediments were taken (0-1 cm or 0-10 cm has the meaning). I would also like to draw your attention to suspicious values ​​for dissolved oxygen. The fact is that the solubility of oxygen decreases with an increase in salinity and decrease in atmospheric pressure. The maximum concentration of DO obtained in the lake with the highest salinity at low atmospheric pressure (at an altitude of about 4.5 km) raises big questions. These values ​​exceed the theoretically possible under such conditions. Please check your device and all the results for validity. Or if you are confident in the data received, then explain it in the manuscript.
And one more small note. A distance of 300 km seems to be too small to analyze the effect of geographical distances if the lakes are not separated from each other by physical geographical barriers. In any case, literature data is worth analysing more seriously, if one exists.
Round 2
Reviewer 3 Report
Dear Authors!
Thank you for your attention to all the comments and for trying to explain your position. In part, I can accept some arguments, but in general I would like to see not a personal (subjective) but a scientific (factual) argumentation of your position. Namely:
(1) Your explanation: Due to the influence of tidal action and wind speed, water flow and biological disturbance, the vertical mixing of coastal surface sediments and overlying water is relatively strong, and the pelagic-benthic coupling is obvious.
This statement is not as obvious as you write. As an example, I will cite the article by Kallistova et al. 2020 (https://link.springer.com/article/10.1007/s00792-020-01185-x), where you can find direct measuremets demonstrating a big difference between parameters of the near bottom water and upper sediments (0 –5 cm) in shallow (less than 1 m) saline (7-140 g/l) rivers (i.e., with water flow and water mixing). There are a lot of publications on the invertebrates inhabiting the sedimnets of these rivers, indicating its biological disturbance, too.
Therefore, your statement requires proof, that is, direct measurements of the parameters of interest in the water column and in the upper layers of sediments.
(2) About the DO measurements:
(i) Your explanation: Most of the lakes in this study are mainly supplied by surface runoff as well as rainfall, but DXC is directly recharged by meltwater from glaciers and snow mountains. Its southern mountains are covered with snow all year round (Figure 1), and the increase in summer temperatures melts the snow and ice in the southern mountains, thus increasing the amount of meltwater inflow into the lake [1]. Therefore, on the one hand, the inflow of meltwater increases the fluidity of the lake water, thus increasing dissolved oxygen [2]; on the other hand, meltwater also lowers the temperature of the lake, thus increasing dissolved oxygen to some extent [3]. And because the samples were collected on the shore, the mixing effect of water and air was more intense, which led to higher dissolved oxygen.
This explanation is unconvincing. Regardless of the way the lake is fed and the degree of dilution of its water with glacial waters, we see the resulting salt content of about 107 g/l, which is very high. This means that the melting glaciers do not dilute the water of the lake so much. Or this factor dilutes the lake water in some other location, but not at the sampling point. I admit that it can dilute surface waters, but not bottom waters. But, by the way, this again calls into question your previous statement about the intensive mixing of the water column and the unification of the parameters of bottom water and surface sediments).
So, in any case, we have a salinity of 107 g/l. For this reason, I cannot accept your explanation for the anomalously high DO values.
(ii) Your explanation: We also compared the dissolved oxygen data of this study with the data of Ranwu Lake, which is a glacial-fed lake in Tibet (Figure 2) [4]. Through comparative analysis, we found that: firstly, the DO of different lake areas in Ranwu Lake was mostly distributed between 6.34-7.38 mg/L, which was similar to the dissolved oxygen values of most lakes we measured; secondly, the dissolved oxygen values in summer were mostly distributed between 6.5-7.2 mg/L, and the DO of QCH directly supplied by glacier meltwater reached 8.14 mg/L, indicating that the increase of melt water plays a positive role on dissolved oxygen content; finally, in all the sites studied of Ranwu Lake, the maximum oxygen content (11.46 mg/L) was also measured at the site RNB (206.00 μS cm-1) with the highest conductivity, which may also indicate that although salinity restricts the dissolved oxygen content to a certain extent, it is also affected by other local factors.
I also cannot accept this explanation and this analogy, since the maximum conductivity given in the cited work (206 μS cm-1) corresponds to 103-144 ppm (=mg/l) depending on the conversion coefficient. It means that fresh waters were analyzed, but not salt waters. This is not your case.
(iii) My own comment. Thank you for indicating the name of the device used to measure DO and other parameters (550A, YSI, USA). I found its Manual (https://www.ysi.com/File%20Library/Documents/Manuals/605348-YSI-550A-Operations-Manual-RevB_001.pdf). It turned out that this device has salt compensation for measurements in the range of salinity up to 70 ppm (=g/l). Hence one question and one statement:
First, answer honestly, did you set the salinity value for the salt compensation at each measurement in the field? If not, then, unfortunately, most of the obtained values are incorrect and cannot be used for analysis.
And secondly, according to the technical characteristics, your device cannot correctly measure the DO content at a salinity above 70 g/l, which means that this value (12.82 mg/l) for the DXC (107 g/l) lake is unambiguously incorrect and cannot be used for analysis.
I will give a Major Revision this time, although I think that my comments are quite serious and require the inclusion of the results of in situ measurements and analyzes, which are impossible to do post factum, since the natural parameters have changed. But perhaps the Authors have the results of these measurements, then they need to repeat their analysis taking correct parameters into account. Otherwise, I'm sure the article should be rejected. Although the studied lakes are of great scientific interest, and I'll be very sorry to reject the manuscript.
Round 3
Reviewer 3 Report
Dear Authors!
The third round of peer review showed me that, unfortunately, while working in the field, you apparently did not understand the specifics of the studied lakes. And apparently there are no experts in the microbial ecology among the authors. This is what your responses to my comments showed me. The fact that you did not attach importance to the model and technical capabilities of the device for measuring parameters in such specific conditions also testify in favor of this. First you did not specify the device at all, then you specified "550A, YSI", then you changed it to "YSI ProDSS". YSI ProDSS seems to be really good when used with the additional electrodes, but for some reason you wrote this model of device only after the reviewer pointed out that it is impossible to get correct data using the 550A. Unfortunately, this sequence of your responses makes me doubt that while working in the field, you took measurements consciously and correctly.
But regardless of the instrument used, you cannot provide evidence that the parameters measured in the bottom water will be correct for the 0-5 cm of sediments. Your explanations about intensive mixing are unconvincing. I would also like to draw your attention to the values ​​of the water content (W in Table 1) in the sediments, which vary from 10 to 60%, but mostly in the range 13-20%. My own and my coleagues' field experience shows that these are rather dense sediments that are not so easy to mix and homogenize with water. I am absolutely sure that the gradient of physico-chemical parameters between the bottom water and the upper layer of sediments must be significant in your lakes.
Finally. On the one hand, you did a great job with interesting objects and idea, but on the other hand, your research was conducted incorrectly. For all my claims, you offered only explanations, which I cannot accept as evidence. Me and my colleagues have extensive experience in field work with saline ecosystems of various nature and chemical composition (both marine and lakes), so I know the specifics of such environments in practice. Indeed, in nature it happens very differently. That is why I asked for the evidence (facts, results, direct measurements) to support the validity of using water parameters in the analysis of sediments. Unfortunately, you have no such evidence. In this regard, I am forced to recommend rejecting this work.